# Factors Affecting Vitamin D Status in Infants

**DOI:** 10.3390/children6010007

**Published:** 2019-01-08

**Authors:** Charles Fink, Rachel L. Peters, Jennifer J. Koplin, Justin Brown, Katrina J. Allen

**Affiliations:** 1Monash University, Faculty of Medicine, Nursing and Health Sciences; Wellington Road, Clayton, VIC 3800, Australia; justin.brown@monashhealth.org; 2Murdoch Children’s Research Institute, 50 Flemington Road, Parkville, VIC 3052, Australia; rachel.peters@mcri.edu.au (R.L.P.); jennifer.koplin@mcri.edu.au (J.J.K.); katrina.allen@mcri.edu.au (K.J.A.); 3University of Melbourne, School of Population and Global Health, Grattan Street, Parkville, VIC 3010, Australia; 4Monash Children’s Hospital, Department of Paediatric Endocrinology and Diabetes, 246 Clayton Road, Clayton, VIC 3168, Australia; justin.brown@monashhealth.org; 5University of Melbourne, Department of Paediatrics, Grattan Street, Parkville, VIC 3010, Australia; 6Royal Children’s Hospital, Department of Allergy and Immunology, 50 Flemington Road, Parkville, VIC 3052, Australia; 7University of Manchester, the Institute of Inflammation and Repair, Oxford Rd, Manchester M13 9PL, UK

**Keywords:** vitamin D, infants, vitamin D supplementation, maternal supplementation, ethnicity, infant formula, breastfeeding, UV exposure, latitude, socioeconomic status

## Abstract

Vitamin D is critical to children’s skeletal development and health. Despite this, the factors which determine vitamin D concentrations during infancy remain incompletely understood. This article reviews the literature assessing the factors which can affect vitamin D status in infancy, including antenatal and postnatal vitamin D supplementation. Observational data supports that dietary intake of vitamin D, UV exposure, and geographic factors contribute significantly to infants’ vitamin D status, but the relationship is unclear regarding genetic variation, ethnicity, and maternal vitamin D status. Randomised controlled trials have compared higher versus lower doses of infant vitamin D supplementation, but no studies have compared infant vitamin D supplementation to placebo and eliminated external sources of vitamin D to fully quantify its effect on vitamin D status. Knowledge gaps remain regarding the factors associated with optimal vitamin D concentrations in infants—including key factors such as ethnicity and genetic variation—and further studies are needed.

## 1. Introduction

Vitamin D is a secosteroid synthesised in the body using sunlight exposure [1]. Without it, normal bone development is impossible, as vitamin D is an essential part of calcium and phosphate metabolism and skeletal mineralisation [2]. This is best demonstrated by rickets, which can be a serious consequence of severe vitamin D deficiency (VDD) and can cause debilitating bone deformities lasting for decades [3].

There is emerging evidence that vitamin D may also have non-osseous health benefits (e.g., immunological) [4], as its receptor can be found in many tissues in the body [5]. New large randomised trials—such as the Vitality trial [6]—are in the process of assessing the effect that vitamin D may have outside of the skeletal system. The results of this trial have not yet been published.

In the general population, vitamin D status varies widely, and understanding the causes of these variations is clinically and scientifically important. The factors which affect vitamin D in adults have been extensively investigated, with the key factors being UV exposure levels (through differences in latitude and sun protection behaviours), genetic and ethnic variation, and skin pigmentation [1,7,8]. In paediatric populations, however, the evidence is not as abundant. Few studies have been carried out in children, however the factors affecting vitamin D status in this age group seem to be similar to those which affect adults [9]. Further research in paediatric age groups are required. 

In infant populations, there is a growing body of published research. Some of the factors which affect adults’ vitamin D concentrations affect those of infants similarly: for example, latitude, skin pigmentation, and seasons. Some factors are important across the lifespan but affect infants in a different way, such as differences in sun exposure behaviours. Other factors that contribue to infant’s vitamin D status are unique to this age group entirely, include maternal vitamin D status and infant feeding practices.

The published literature includes review articles of the determinants of vitamin D status in adult populations [1,7,8], but despite the importance of vitamin D in infancy, the factors affecting vitamin D status in this age group has not been similarly reviewed.

The purpose of this article is to evaluate the factors affecting vitamin D status in infants across a range of levels. Strategies to prevent infant VDD include infant and maternal vitamin D supplementation, and studies addressing these are critically reviewed. The latter can be divided into antenatal supplementation, aiming to improve vitamin D status at birth; and postnatal supplementation, aiming to improve dietary intake of vitamin D. The effect of UV exposure on vitamin D status, including seasonal variation and geographical factors such as latitude, are included in this review. Additionally, demographic factors such as ethnicity and socioeconomic status are reviewed.

## 2. Background

### 2.1. Physiology

The production of vitamin D involves primarily the skin, the liver, and the kidneys [1]. In the skin, exposure to UV radiation catalyses the conversion of 7-dehydrocholesterol to previtamin D3, which is then isomerised with heat to vitamin D3. The primary storage form of vitamin D is 25-hydroxyvitamin D (25(OH)D), which is formed after hydroxylation in the liver. 25(OH)D is hydroxylated in the kidneys to form calcitriol (or 1,25-dihydroxyvitamin D), which is the biologically active form of vitamin D [1].

### 2.2. Definitions

Although there is variation in international guidelines as to what defines vitamin D deficiency, Australian and European guidelines define vitamin D deficiency as serum 25(OH)D concentration less than 50 nmol/L [10,11,12]. British, American, and international consensus guidelines differ slightly, defining sufficiency as greater than 50 nmol/L [13,14,15]. For the purpose of this article, deficiency will be defined as less than 50 nmol/L, in line with Australian and European guidelines. 

Units of measurement used internationally to describe amounts of vitamin D include:Doses: 1 microgram (µg) = 2.5 nanomoles (nmol) = 40 international units (IU)Concentrations: 1 ng/mL = 1 µg/L = 2.5 nmol/L

## 3. Search Methodology

Studies assessing the determinants of vitamin D in infants were searched for using Ovid MEDLINE and Cochrane Library on 13 August 2018. The search terms (matched to subject heading (MeSH) when possible) were “vitamin D” and “infant” and (“factor*” or “determinant*” or “aspect*”). The search yielded 314 publications and the abstracts and titles of the publications were assessed for relevance and whether they met the eligibility criteria. Studies were excluded if they focused on vitamin D deficiency causing illness (rather than factors causing vitamin D deficiency), their study population did not include children less than 2 years of age, or they weren’t written in English. 11 studies from the search were included in this review, and the reference lists of included articles were searched for additional articles. As each study assessed different factors in participants of varying ages in infancy, this narrative review is organised by factor, with the findings of each study assessing that factor collated.

## 4. Maternal Vitamin D Status

Neonatal vitamin D status is linked to maternal vitamin D, as 25(OH)D is understood to be transferred across the human placenta to the foetus, as observed in studies of rats [16]. Consequently, Australian prevalence rates of VDD at birth reflect the rates of maternal deficiency [17]. This has also been observed internationally: in the US [18], Germany [19], Turkey [20], and Pakistan [21].

### 4.1. Antenatal Maternal Supplementation

It has been hypothesised that maternal vitamin D supplementation during gestation may prevent infant VDD. A meta-analysis of 43 randomised trials has demonstrated that antenatal maternal supplementation raises cord blood 25(OH)D [22]; however, as the half-life of 25(OH)D is approximately 3 weeks in serum [23], 93.75% of this vitamin D at birth will be catabolised by 3 months of age. 

Merewood et. al completed a cohort study in 177 infants in the US and reported that although maternal serum 25(OH)D was correlated with infant vitamin D at birth [24], this correlation was no longer observed at 4 months of age [25]. The meta-analysis described above observed that maternal vitamin D supplementation was associated with increased 25(OH)D concentrations in their offspring [22]. Some of the studies included in the meta-analysis included a placebo comparison, while others compared varying doses of vitamin D (range 200-600 IU/day or equivalent), including regular and bolus dose regimens. Although an association was observed at birth, the follow-up periods of these studies do not extend past 6 months of age. The trial with the longest follow-up period found that the beneficial effect granted by antenatal supplementation at birth was reduced by age 2 months and did not extend to 4 or 6 months [26]. In our cross-sectional study of 563 12-month olds in Melbourne, Australia, we found that self-reported antenatal maternal supplementation was not associated with a reduced risk of VDD at age 12 months [27]. These findings suggest that any benefit granted by antenatal maternal vitamin D supplementation does not extend beyond 6 months.

## 5. Dietary Intake

The natural vitamin D content of the typical Western diet is low without fortification [1,28]; dietary intake of vitamin D in Australian adults is estimated to be only ≈80–120 IU/day [29], 20–30% of the estimated average requirement [10]. In the US, due to increased fortification, dietary intake in adults is higher (≈144–288 IU/day) [30], but remains only 24–48% of the recommended dietary allowance [14]. In the first 4 months of life, an infant’s diet consists almost entirely of breastmilk and/or infant formula. Solids are commenced around 4–6 months of age [31], with 90% of infants introduced to solid foods before 7 months of age [32]. Despite the many benefits of breastmilk, maternal vitamin D is poorly transferred in human milk. The average vitamin D content of breastmilk has been observed to range from 25–124 IU/L [33,34]. This content reflects maternal vitamin D status, and may reach a maximum of 80 IU/L with standard supplementation doses [33]. The extent to which the vitamin D content of breastmilk drops in cases of maternal VDD is not known.

### 5.1. Postnatal Maternal Supplementation

It has been hypothesised [35] that with higher maternal supplementation doses, the vitamin D content of breastmilk can be raised, and infant and maternal VDD can be prevented with one intervention—postnatal maternal supplementation. Several randomised controlled trials (RCTs) have assessed the efficacy of this practice, using both regular and bolus dosing regimens (dose range 250–4000 IU/day or equivalent), and have observed it to raise both infant and maternal vitamin D status [35,36,37,38]. However, the follow-up period of these studies only extends to 7 months of age at the most [35]. Longer RCTs are required to assess the long-term benefit of this practice.

Additionally, the benefit of this practice is dependent on the length of exclusive breastfeeding. As infants grow older, their diets become less liquid-based [32]. Thus, the benefit of postnatal maternal supplementation may decrease as the infant begins the transition to a solids-based diet.

### 5.2. Formula Intake

In Australia, infant formula is fortified with 360–520 IU/L of vitamin D [10]. This level of fortification is similar to other countries, such as the Netherlands [39], the US and Canada [40]. This provides approximately enough vitamin D for an infant, but only if the infant is exclusively formula-fed. According to infant feeding guidelines [31] and weight growth charts [41], the median infant will drink up to 1 litre of formula per day (Table 1).

The relationship between dietary vitamin D intake and vitamin D status has been found to be one of the more critical factors in infants. As infant formula contains vitamin D, increased formula intake would be expected to be associated with a reduced risk of vitamin D deficiency. We previously demonstrated that mixed formula- and breastfeeding and exclusive formula-feeding were associated with a reduced risk of VDD at 12 months (*n* = 563) compared to exclusive breastfeeding (adjusted odds ratio (aOR) 0.30 (95% confidence interval (CI) 0.12–0.76, *p* = 0.01) and aOR 0.20 (95% CI 0.11–0.36, *p* < 0.001), respectively) [27]. Mixed feeding represents more infant formula intake than exclusive breastfeeding but less than fully formula-fed diets. This was associated with a reduced risk of vitamin D deficiency compared to exclusive breastfeeding but exclusive formula-feeding reduced the risk of deficiency further. Carpenter et al. reported similar findings: formula-feeding in the first 6 months of life was correlated with higher vitamin D concentrations (regression coefficient = 5.13, *p* = 0.008) [42]. These findings contribute to observational evidence linking exclusive breastfeeding with vitamin D deficiency.

## 6. Infant Supplementation

Multiple studies have assessed the effect of vitamin D supplementation on serum 25(OH)D in infants. Studies of high-dose (e.g., 100,000 IU) dosing regimens (referred to as bolus dosing) have found that although effective in raising 25(OH)D concentrations in the short-term, the effect of these doses only lasts between 1.5–4 months [43,44]. Strategies similar to this, however, may be useful in populations where compliance to daily supplementation is low.

RCTs which assessed daily drops of 400 IU in the first year of life (the dose recommended in clinical practice guidelines [12,13,45,46]) are summarized in Table 2 (design) and Figure 1 (results). The results of these trials indicate that infant vitamin D supplementation raises serum 25(OH)D concentrations.

Rueter et al. (*n* = 195) administered daily doses of 400 IU vitamin D from 1–6 months of age, and compared the rise in serum 25(OH)D to a placebo group, in a double-blind RCT [47]. The researchers observed that the mean serum 25(OH)D of the intervention group at 6 months of age was higher than that of the control group (93.1 nmol/L vs. 82.0 nmol/L, *p* = 0.2, Figure 1). They did not observe a statistically significant difference between the groups with regards to their primary outcome, the development of eczema and doctor-diagnosed wheeze at 6 months of age. This trial’s major strength lies in its RCT design and use of a placebo comparison. However, the serum 25(OH)D of these participants may not have been caused by group allocation alone and may have been affected by infant formula intake. Although participants who ingested greater than 1L of formula per day were advised to discontinue the intervention (either placebo or vitamin D drops), their results were included in the analysis, as per intention-to-treat procedures. This should be compared with a per protocol analysis so that the rise attributable to the intervention alone can be seen. Additionally, participants who ingested less than this amount (reported as 56% of the placebo group at 6 months of age, not reported for the intervention group) were also included and not analysed separately. These factors may account for the large rise in serum 25(OH)D seen in the placebo group from age 3 months to age 6 months (22.8 nmol/L); a greater rise than that seen in the intervention group (9.9 nmol/L). Data on compliance to the intervention were not reported, which is important to assess the efficacy of the intervention. Finally, mothers recruited for the trial were screened for vitamin D deficiency in the third trimester of pregnancy and excluded if found to be deficient, which limits generalisability of this study to the wider population. The results of this trial suggest that vitamin D supplementation may raise serum 25(OH)D concentrations, but in order to assess the effect of vitamin D supplementation alone, the analysis needs to account for formula intake as an external source of vitamin D.

Alonso et al. administered daily doses of 400 IU vitamin D from 1–12 months of age and compared their mean serum 25(OH)D to a control group which received no treatment, without blinding [48]. In this trial, formula-fed infants were not excluded (representing an external source of vitamin D), meaning that any effect detected on the outcome may not be due to the supplementation alone. In their results, they found very similar vitamin D concentrations in both groups at 12 months of age (Figure 1).

Ziegler et al. completed a double-blind RCT and restricted their sample to exclusively breastfed infants alone [49]. The intervention period extended to 9 months of age (although follow-up continued until 12 months of age). Their results indicate that 400 IU administered daily increased serum 25(OH)D concentrations, but its effectiveness when continued until 1 year of age was not tested by this trial. Additionally, without a placebo comparison (which is unethical in the USA due to supplementation being the standard of care), how much of the change in 25(OH)D concentration is due to the intervention is unknown.

Gallo et al. completed a double-blind RCT, restricted their sample to exclusively breastfed infants alone, and administered the intervention for the first year of life in its entirety [50]. They found that 400 IU also raised 25(OH)D concentrations, but like Ziegler et al., the researchers were unable to compare their intervention to placebo. 

Collectively, these studies show that vitamin D supplementation is associated with raised serum 25(OH)D concentrations in infants. Without a placebo comparison, combined with exclusion of external sources of vitamin D, the rise in serum 25(OH)D which is attributable to the supplementation alone is unclear.

Each of these trials used a different method of quantifying 25(OH)D: chemiluminescent assay (Rueter et al. and Alonso et al.), radioimmunoassay (Ziegler et al.), and liquid chromatography tandem mass spectrometry, including the separation and quantification of 3-epi-25(OH)D3 (Gallo et al.). Only Rueter et al. and Gallo et al. documented the use of external quality assessment systems (VDSP [51] and DEQAS [52] respectively), which may prevent the accurate comparison of these results.

## 7. UV Exposure and Seasonal Variation

Since infants are at a higher risk of sunburn [53], it is recommended that they be kept out of direct sunlight, especially in the first 12 months of life [54]. Australian guidelines suggest for infants to be kept in sun-protected carriages, and with most skin covered by blankets or clothing [54]. Other countries also have similar recommendations for infants [55,56]. Sun protection has the secondary effect of preventing the baby from producing vitamin D cutaneously.

Measures of UV exposure vary between studies; however, a positive correlation has been shown between UV and vitamin D status [25,27,57]. Some studies use surveys of sun exposure behaviour alone to assess UV exposure, while others use ambient UV radiation levels. Measurements based on surveys are subject to recall bias, but they measure the individual more closely than ambient UV levels do. Merewood et al. observed that in 4-month-olds, less than 1 day/week with 10 or more minutes spent outside was associated with increased risk of deficiency (aOR 0.12, 95% CI 0.02–0.66, *p*-value not reported) [25]. In contrast, Grant et al., in a study of 353 New Zealand infants aged from 6 months to 2 years, found no significant relationship with hours spent outdoors or wearing sunscreen or a hat [57]. The researchers did observe, however, that wearing long-sleeved tops and pants outdoors increased the risk of VDD (relative risk (RR) = 7.57, 95% CI 1.33–26.87). We utilised ambient UV levels to estimate individual UV exposure and observed that higher UV exposure was strongly correlated with reduced risk of VDD in 12-month-olds (aOR = 0.53, 95% CI 0.43–0.66, *p* < 0.001) [27]. A final method for measuring personal UV exposure is personal dosimetry, which is a device worn by the participant that measures the UV exposure directly received by a participant. Rueter et al. utilised personal dosimeters to estimate the amount of UV radiation their participants were exposed to from 1–3 months of age [47]. The researchers found no correlation between UV exposure at ages 1-3 months and serum 25(OH)D concentration in their placebo group at either age 3 months (*p* = 0.52) or at age 6 months (*p* = 0.70). These results are in contrast to the findings of the above studies which used less personalised measurements of UV exposure, and may suggest that UV exposure and vitamin D status in infancy are not as closely linked as otherwise thought. The absence of an association between UV exposure and vitamin D status may also be affected by the low amounts of UV exposure in the first three months of life, or by the low numbers of participants in the analysis group (*n* = 38). Further studies using accurate personalised dosimetry of infants coupled with serum 25(OH)D measurements in the first year of life with larger sample sizes are required to better explore this association.

It has been observed that 25(OH)D concentrations in infants are lowest in spring, probably due to the low UV levels in winter. Siafarikas et al., in a large (*n* = 3481) cross-sectional study of infants under 12 months old, observed that vitamin D concentrations were lowest in March (i.e., after winter in the northern hemisphere) (*p* = 0.01) [58]. These findings were corroborated by a large Chinese cross-sectional study [59].

## 8. Geographic Factors

Geography can directly affect vitamin D status through latitude. Modelling has shown that, in the same conditions (season, skin pigmentation, skin exposed, and time spent outdoors), the amount of vitamin D produced by a patient in Darwin (latitude 12° S) is eight times greater than that produced by one in Hobart (latitude 42° S) [60]. The study by Siafarikas et al. spanned multiple latitudes in East Germany and found that mean serum 25(OH)D concentrations were higher in the south than in the north (78 nmol/L at 54.1° N, 83 nmol/L at 51.5° N, and 88 nmol/L at 50.6° N) [58].

Geography can also affect vitamin D status through local health policy: some international guidelines (e.g., the US, the UK, Europe, Canada) recommend 400 IU of vitamin D supplementation for infants under 12 months old [12,13,45,46]. Current Australian guidelines only recommend supplementation to exclusively breastfed infants with one or more other risk factors for deficiency [9]. These include dark skin (Fitzpatrick types V and VI), reduced UV exposure (e.g., from cultural practices or extreme southerly latitudes), or medical conditions affecting fat metabolism (e.g., cystic fibrosis).

## 9. Genes and Ethnicity

Ethnic heritage can affect one’s vitamin D status, through a combination of genotypic and phenotypic (e.g., skin pigmentation) factors. Gene changes in vitamin D bioavailability pathways have been observed to affect vitamin D status, such as through variation in the gene which codes for the vitamin D binding protein (DBP). As vitamin D is lipid-soluble, DBP is needed for its transport in the bloodstream [61]. When bound to DBP, vitamin D is protected from catabolism and renal filtration but is also biologically inactive [62]. It is suggested that the function of DBP is to maintain stable concentrations of vitamin D when vitamin D production is variable (e.g., during winter) [62,63].

An observational study (*n* = 2085 adults) found that African Americans were more likely to have lower DBP concentrations than Caucasians, and the researchers confirmed a higher proportion of alleles linked with lower DBP concentrations in African Americans [63]. Although African Americans had lower total 25(OH)D concentrations (due to increased skin pigmentation), both groups had similar free 25(OH)D concentrations, which was attributed to the differences in DBP. The authors speculated that these alleles may have evolved in different ethnicities in order to compensate for reduced vitamin D production (due to darker skin). Since DBP-bound vitamin D is biologically inactive, less DBP allows for a higher free-to-total vitamin D ratio and for patients with less total vitamin D to maintain normal concentrations of free vitamin D [63].

The findings of this study have been discussed by several peers in the field who suggested that the results may have been affected by the monoclonal antibody assay used to quantify DBP in these participants [64]. A more recent study has observed racial differences in DBP levels only when using this monoclonal assay [65]. When other assays were used, no significant racial difference was noted. The researchers of the more recent study suggest that the monoclonal immunoassay may underestimate DBP concentrations in some genotypes, such as those seen in African-American populations.

The relationship between ethnicity and vitamin D has also been explored in infant populations. The findings in adults may not be generalisable to infants, as aspects which may modulate this relationship, such as infants’ sun exposure behaviours, differ to adults’. Grant et al. observed an increased risk of VDD in infants of Pacific ethnicity and Maori ethnicity compared to those of European ethnicity (RR 7.60, 95% CI 1.80–20.11 and RR 4.30, 95% CI 1.01–13.78, respectively) [57]. The authors suggested that this difference was in part, but not solely, due to skin pigmentation, as Maoris have similar levels of skin pigmentation to those of Pacific ethnicity. They suggest that this disparity may be due to cultural and lifestyle differences. However, the difference in these relative risks is likely not statistically significant, as the confidence intervals overlap [57]. Carpenter et al., in a study of 750 infants aged between 6 months and 3 years, found that Caucasians had higher 25(OH)D concentrations than Hispanic and African American infants, respectively (univariate logistic regression: with African Americans as baseline, regression coefficient = 5.3 and 8.3 for Hispanics and Caucasians, respectively, *p* < 0.002) [39]. After adjusting for skin pigmentation, however, the effect that ethnicity had on serum 25(OH)D was not statistically significant. 

The relationship between ethnicity and vitamin D status may be even more multifaceted than demonstrated by these studies. Infant formula intake rates vary across ethnic and cultural groups [66], and may have modulated this relationship, however these studies did not stratify their analysis by breastfeeding status. Similarly, cultural differences in clothing worn and time spent outdoors could have affected the UV exposure of these babies and their mothers, which may also have contributed to the differences seen.

In summary, ethnicity may affect vitamin D status through skin pigmentation, allelic variation, and lifestyle differences across cultures, but further studies with thorough assessments of all these factors are required.

## 10. Individual factors

### 10.1. Socioeconomic Status

The relationship between socioeconomic status and vitamin D status has been assessed in several studies [57,67]. Camargo et al. utilised the New Zealand Deprivation Index [68] and found that the most affluent participants had higher mean 25(OH)D concentrations than the poorest (median 53 nmol/L vs 36 nmol/L, *p* = 0.001 for trend) [67]. This trend was not statistically significant on multivariate analysis, however (aOR 0.97, 95% CI 0.92–1.03, *p* = 0.30). In contrast, Grant et al. used a more direct measure of socioeconomic status: household expenditure-to-income ratio [57]. A ratio of < 0.34 was used as the baseline: i.e., spending less than $1 for every $3 earned. Infants in households with a ratio greater than 0.34 were at higher risk of VDD, after adjusting for ethnicity (relative risk = 7.98, 95% CI 4.59-13.55). The researchers suggest that higher expenditure-income ratios in those of Pacific ethnicity may be a reason why the researchers observed a higher risk of VDD in Pacific participants in their study (see above).

These findings suggest that there may be a relationship between lower socioeconomic status and higher risk of vitamin D deficiency; and that socioeconomic differences may explain some of the variation in vitamin D deficiency prevalence between ethnicities. Further research into this relationship is warranted.

### 10.2. Obesity and Physical Activity

As vitamin D is a fat-soluble vitamin, an individual’s body mass index has been shown to affect vitamin D status. Increased adiposity has been observed to be correlated with decreased serum 25(OH)D concentrations in both adult [69] and paediatric populations [70]. No studies have explored this relationship in infancy. Additionally, a newborn’s vitamin D status might be affected by their mother’s adiposity, and this association should be explored in future studies.

Similarly, increased physical activity has been observed to correlate with increased serum 25(OH)D concentrations in adult populations [71,72]. Physical activity as measured by these studies is not possible in infancy, but again, it may affect an infant’s vitamin D status via maternal vitamin D status. Research is required to assess the relationship between maternal exercise levels and their relationship with infant vitamin D levels, adjusted for the potential confounder of increased UV exposure.

## 11. Conclusion

In summary, there are a number of observational studies which have studied factors affecting vitamin D status in infants, but this evidence is not conclusive. Observational data has shown significant associations between dietary intake, UV exposure, latitude, seasonal variation, and infants’ vitamin D status. Although some associations between genetic variation, ethnicity, socioeconomic status, and vitamin D status have been reported, these are not yet completely understood.

Randomized trials assessing these relationships are not possible because of the nature of the exposures, with the exception of supplementation. The long-term effect of maternal vitamin D status on infant vitamin D status is unclear, and further trials with longer follow-up periods are required. Finally, infant supplementation trials should include a placebo group as comparison as well as excluding external sources of vitamin D in order to truly assess the effect of supplementation. Further studies are warranted for a holistic clinical understanding of vitamin D in this critical age group.

## Figures and Tables

**Figure 1 children-06-00007-f001:**
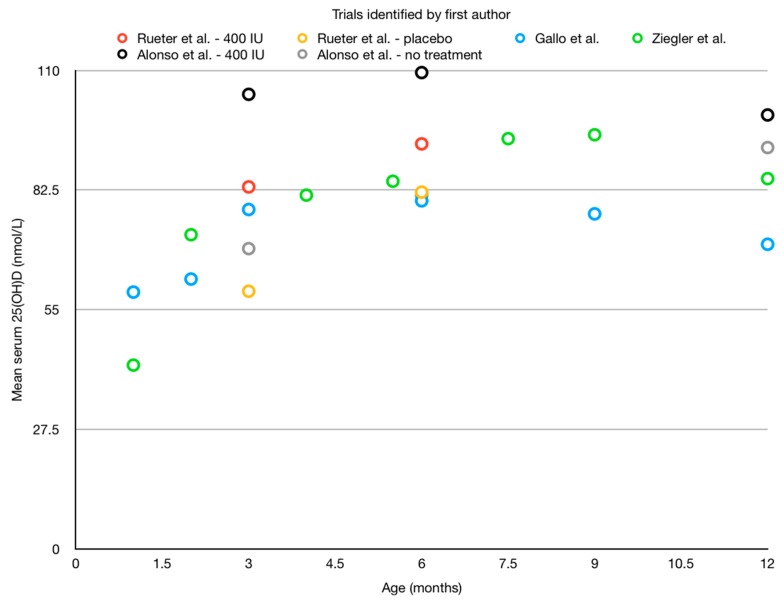
Mean serum 25-hydroxyvitamin D (25(OH)D) measurements in randomized controlled trials (RCTs) assessing 400 IU/day vitamin D supplementation in 1–12-month-old infants. These measurements are the mean serum 25(OH)D at each timepoint in each experimental group which received 400 IU of vitamin D supplementation in these studies.

**Table 1 children-06-00007-t001:** Estimates of vitamin D intake in fully formula-fed infants.

Age (Months)	Formula Intake (mL/kg/day)	Median Weight (kg)	Total Formula Intake (mL/day)	Estimated Vitamin D Intake (IU/day)
0–3	150	6 (3 months)	900	324–468
3–6	120	8 (6 months)	960	345–499
6–12	100	10 (12 months)	1000	360–520

Adapted from National Health and Medical Research Council infant feeding guidelines [30] and World Health Organisation growth charts [38]. IU = international units.

**Table 2 children-06-00007-t002:** Study design, participants, and details of RCTs assessing 400 IU/day vitamin D supplementation in 1–12-month-old infants.

First Author	Dose (IU)	Comparison (IU)	Intervention Duration (months old)	Blinding	Sample Size of 400 IU Group		Country
					Randomisation	Analysis	
**Rueter, K. [47]**	400	placebo	1–6	double	98	86 *	Australia (35°S)
**Alonso, A. [48]**	400	no treatment	1–12	none	41	30	Spain (43°N)
**Ziegler, E.E. [49]**	400	200,600,800	1–9	double	60	30	USA (41°N)
**Gallo, S. [50]**	400	800,1200,1600	1–12	double	39	29	Canada (45°N)

* this number includes infants who discontinued the intervention, and it is unclear if it includes infants who attended the 6-month endpoint but did not have their 25(OH)D measured. IU = international units.

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
