# Peer review of "Factors Affecting Vitamin D Status in Infants"

_children, 2019, doi:10.3390/children6010007_

Round 1

Reviewer 1 Report

1.  There are some areas that could use updated references (e.g. Ref #7, 15, 32).

2.  Line 37-39: This study contains little synthesis of the information. Please provide more information about the results of this study.

3.  Line 40-48: What about children? Is there any evidence on factors affecting vitamin D in children?

4.   One of the first things that should be clear in the introduction of a review is a clear description of the evidence gap that the review is filling- I unfortunately could not find such a justification for the work presented here. This should be addressed at the close of the introduction.

5. Search strategy should be presented in a Table identifying search terms, inclusion and exclusion criteria, databases and search findings.

6. The results/descriptive section starting on line 76 could use additional synthesis. For example, would be helpful to discuss this hypothesis further (Line 111-113)? This paragraph should be presented with major details (Line 128-135). Why would mixed formula and breastfeeding be associated with a reduced risk of VDD?

7. It is not clear to me whether this review focused on the Australian context or different cultural context. Most of the detail described can tell in Australia (e.g. Line 121-124). Is there any evidence from other contexts?

Reviewer 2 Report

In this study, the authors have written a narrative review describing the factors affecting the vitamin D status of infants. Thank you for the opportunity to review this paper. The study in general was well written and provides a nice overview of the current knowledge around this topic. 

Some specific comments and suggestions are listed below:

•   Are the authors aware of the VITAL trial where infants were supplemented daily (or not) with 400 IU vitamin D3 for 6 months, with blood 25(OH)D levels measured at 3 and 6 months, and included a placebo control? See Rueter et al 2018. J Allergy Clin Immunol. These researchers also measured sun exposure by these babies. These new data should be considered in the infant vitamin D supplementation (6) and UV exposure (7) sections.

•   Consideration or discussion of the effects of maternal and infant adiposity on serum 25(OH)D levels is needed especially as there are known inverse relationships in adult and child populations.

•   Similarly, there is some evidence that physical activity might modify serum 25(OH)D, although this might be more relevant for maternal vitamin D status.

•   The introduction should contain a brief review of the pathways whereby vitamin D is produced in skin and systemically following UV exposure, and also through ingestion of dietary vitamin D, particularly with reference to the infant. A figure could be used here.

•   Is the half-life of 25(OH)D different in infants and children to adults? If so, this could affect the length of time maternally-derived 25(OH)D lasts in the newborn.

•   Is there any evidence that vitamin D (in utero) is converted to 25(OH)D and/or 1,25(OH)2D by the foetus? Is the assumption that babies are born with the capacity to make their own vitamin D from birth through skin exposure to UV light, or increase their levels via the diet, but has this been tested?

•  In the sections describing vitamin D supplementation, the range of doses and how supplementation was administered should be discussed (Sections 4.1, 5.1).

•   Take care throughout when describing vitamin D status that when referring to 25(OH)D levels or concentrations to use the correct metabolite (e.g. line 133 – ‘higher vitamin D concentrations’, should be ‘higher 25(OH)D concentrations’).

•   The methods used to generate the data presented in Figure 1 should be discussed, especially as you are comparing across 4 different studies. Did all use methods that had been standardized using the VDSP?

•   In Sections 7 and 8, please state the location of where the study by Siafarikas et al was done.

•   Is there any effect of altitude on infant vitamin D status?

•   What about difference in cultural practices around the world that might limit or promote sun exposure by mothers and their babies?

•   Care should be taken when discussing ref 55 (Powe et al, line 219) as the method used to measure VDBP has been widely critiqued, resulting in multiple letters to the editor regarding this publication. This issue should be discussed, along with whether there are any more recent studies that have confirmed these findings (hopefully done by using better assay).

•   Do studies assessing ethnicity or SES upon infant vitamin D status adjust for breast-feeding/formulae feeding?

Round 2

Reviewer 1 Report

No further comments. Thanks for revising the paper.

Reviewer 2 Report

The authors have addressed all of my comments.